# Ph-Positive B-Cell Acute Lymphoblastic Leukemia Occurring after Receipt of Bivalent SARS-CoV-2 mRNA Vaccine Booster: A Case Report

**DOI:** 10.3390/medicina59030627

**Published:** 2023-03-21

**Authors:** Shy-Yau Ang, Yi-Fang Huang, Chung-Ta Chang

**Affiliations:** 1Department of Emergency Medicine, Far Eastern Memorial Hospital, No.21, Sec. 2, Nanya S. Rd., Banciao Dist., New Taipei City 22056, Taiwan; 2Department of General Dentistry, Chang Gung Memorial Hospital, Linkou 33305, Taiwan; 3School of Dentistry, College of Oral Medicine, Taipei Medical University, Taipei 11031, Taiwan; 4Graduate Institute of Dental and Craniofacial Science, College of Medicine, Chang Gung University, Taoyuan 33302, Taiwan; 5Graduate Institute of Medicine, Yuan Ze University, Taoyuan 32003, Taiwan

**Keywords:** mRNA COVID-19 vaccine, bivalent, acute lymphoblastic leukemia, ALL, anti-spike protein immune response

## Abstract

The coronavirus disease 2019 (COVID-19) pandemic is a universal emergency public health issue. A large proportion of the world’s population has had several spike antigen exposures to severe acute respiratory syndrome coronavirus 2 (SARS-CoV-2) infections and/or COVID-19 vaccinations in a relatively short-term period. Although sporadic hematopoietic adverse events after COVID-19 vaccine inoculation were reported, there is currently no sufficient evidence correlating anti-spike protein immune responses and hematopoietic adverse events of vaccinations. We reported the first case of Ph-positive B-cell acute lymphoblastic leukemia (ALL) occurring after a bivalent mRNA COVID-19 vaccine inoculation. The otherwise healthy 43-year-old female patient had a total of six spike antigen exposures in the past 1.5 years. Informative pre-vaccine tests and bone marrow study results were provided. Although the causal relationship between bivalent vaccinations and the subsequent development of Ph–positive B-cell ALL cannot be determined in the case report, we propose that anti-spike protein immune responses could be a trigger for leukemia. Clinicians must investigate the hematopoietic adverse events closely after COVID-19 vaccinations. Further pre-clinical studies to investigate the safety of bivalent mRNA COVID-19 vaccine are required.

## 1. Introduction

In the era of the coronavirus disease 2019 (COVID-19) pandemic, more than 750 million people have been infected globally, with over six million associated deaths [1]. Herd immunity was recommended as one of the best resolutions for this severe acute respiratory syndrome coronavirus 2 (SARS-CoV-2) crisis. Several vaccines for preventing SARS-CoV-2 infection were developed, such as Oxford/AstraZeneca COVID-19 vaccine, Pfizer BioNTech (BNT162b2) vaccine, Moderna COVID-19 (mRNA-1273) vaccine, and Novavax (NVX-CoV2373) vaccine. Although they have shown effectiveness in reducing COVID-19-related mortality and morbidity [2,3,4,5], many concerns about the safety of these vaccines were generated regarding their novel genetic technology and materials [6]. Sporadic hematopoietic adverse events were reported [7,8,9,10]. The bivalent formulations of BNT162b2 (Pfizer-BioNTech) and mRNA-1273 (Moderna) COVID-19 vaccines, which can induce remarkably high levels of anti-spike protein antibodies, have demonstrated their significant efficacy against the Omicron variant and acceptable safety in clinical practice [11]. They are recommended as the first choice of COVID-19 vaccine boosters to protect against the infection of SARS-CoV-2 variants worldwide.

Acute lymphoblastic leukemia is a hematologic malignancy with a heterogeneous cytogenetic appearance and prognosis [12]. It is an uncommon disease with a low incidence rate of about 1.7 per 100,000 persons per year in the United States [13]. The genetic diseases, such as Fanconi anemia, Klinefelter syndrome, Bloom syndrome, neurofibromatosis, ataxia-telangiectasia, and Down syndrome were reported to be contributed to the development of acute lymphoblastic leukemia [14,15,16,17,18,19]. Other risk factors of acute lymphoblastic leukemia include old age (>70 years), chemotherapy, radiation exposure, Epstein-Barr virus, and human immunodeficiency virus infection [20,21,22]. Although tyrosine kinase inhibitors targeted therapy significantly improved the outcome of patients with Philadelphia (Ph) chromosome–positive acute lymphoblastic leukemia, poor prognosis was reported in patients with relapsed and refractory Ph-positive acute lymphoblastic leukemia [23]. 

Many researchers have provided strong support for the safety and benefits of COVID-19 vaccination [2,3,4,5,11,24], but several authors implied immune cell stimulation might be the trigger of the hematologic adverse events after COVID-19 vaccination [7,8,9,10,25,26]. An intense vaccination campaign for herd immunity to against SARS-CoV-2 infection is mandatory, however, it is a continuous process for collecting the safety data of these vaccines, especially the current worldwide used booster bivalent formulations of BNT162b2 (Pfizer-BioNTech) and mRNA-1273 (Moderna) COVID-19 vaccines. Even if such sporadic events can happen coincidentally, they should be reported to shed light on the major safety concerns about these bivalent mRNA COVID-19 vaccines.

Herein, we report the first case of a patient that developed acute lymphoblastic leukemia after a bivalent COVID-19 booster vaccination with mRNA-1273 (Moderna). 

## 2. Case Report

A 43-year-old woman with insignificant medical history received a booster dose (0.5 mL) of the bivalent (Omicron BA.4/BA.5–containing) mRNA-1273 COVID-19 vaccine on 3 January 2023. One day after this injection, she felt ill with dizziness, mild dyspnea, and general malaise. In the following days, her dizziness persisted and the dyspnea became more severe, so she presented to our emergency department on the fifth day after vaccination. At our emergency department, neither fever nor respiratory symptoms were observed. Her SARS-CoV-2 antigen rapid test was negative. She had tachycardia (119 beats/min) with normal blood pressure. There was no abnormal bleeding, petechia, or ecchymosis detected.

The patient received four COVID-19 vaccinations before the last injection, including two doses of adenoviral vector-based vaccines (Oxford/AstraZeneca), a half-dose of monovalent mRNA vaccine (Moderna), and a protein vaccine (NovaVax) on 4 June 2021, 31 August 2021, 15 January 2022, and 15 July 2022, respectively. There was no remarkable discomfort after these inoculations. She had a SARS-CoV-2 infection with only minimal symptoms on 19 August 2022 and recovered in two weeks without any sequelae. A general blood test, abdominal sonography, and low-dose chest computed tomography (CT) for health examination were done on 12 October 2022. The white blood cell count, red blood cell count, hemoglobin, hematocrit, MCV, MCH, MCHC, and platelet were 5730/μL, 4,110,000/μL, 12.3 g/dL, 37.3%, 94.5 fL, 28.3 pg, 34.6 g/dL, and 202,400/μL, respectively. In the differential count of white blood cell, neutrophil, lymphocyte, monocyte, eosinophil, and basophil were 54.2%, 37.9%, 5.8%, 1.5%, and 0.6%, respectively. The abdominal sonography showed no splenomegaly. 

At our emergency department, the white blood cell count, red blood cell count, hemoglobin, hematocrit, MCV, MCH, MCHC, and platelet were 46,390/μL, 2,530,000/μL, 6.8 g/dL, 20.9%, 82.6 fL, 26.9 pg, 32.5 g/dL, and 48,000/μL, respectively. In the differential count of white blood cells, neutrophil, lymphocyte, monocyte, eosinophil, basophil, and blast were 1.0%, 9.0%, 0.0%, 0.0%, 0.0%, and 90.0%, respectively. The microscopic examination of a Giemsa-Wright stained peripheral blood smear revealed almost all the white blood cells on the visual field were blasts and these large white blood cells presented with irregular/clefting nuclei, coarse, clumped chromatin, and occasional nucleoli and high nuclear-to-cytoplasmic ratio (Figure 1). The results of C-reactive protein, lactate, and LDH were 1.455 mg/dL, 2.28 mmol/L, and 240 U/L, respectively. Splenomegaly with tiny splenic infarct, and no enlargement of lymph nodes were shown on the contrast-enhanced CT of the abdomen (Figure 2). 

Two days later, the bone marrow biopsy and aspiration studies were conducted. A 1.2 × 0.2 × 0.2 cm brown, hard, and core-like bone marrow specimen was obtained from the left iliac crest of the patient. The specimen was fixed in formalin subsequently. Microscopically, it shows hypercellular marrow with a cellularity of more than 90%. Most marrow spaces and hematopoietic components are replaced by small to medium-sized primitive round blue cells. Immunohistochemically, the lesion cells are positive for CD34 and Terminal deoxynucleotidyl transferase (TdT); negative for CD117 and myeloperoxidase. CD3 shows only scattered positivity, and CD20 stains on more cells (about 20% of the nucleated cells) than CD3. 

The bone marrow aspiration showed 68% blastic infiltration. The immunophenotype characterization with the use of flow cytometry demonstrated moderate cytoplasmic CD79a (Figure 3a), CD19 (Figure 3a), and CD34 (Figure 3b) expression with negative cytoplasmic myeloperoxidase and CD3. It was compatible with B-cell precursor acute lymphoblastic leukemia. In B-cell precursor acute lymphoblastic leukemia diagnostic panel, the specimen showed brightness of CD58 (Figure 3c) and CD10 (Figure 3d) with moderate expression of CD34, CD19, CD66c, CD38, TdT (Figure 3e), CD24; dim CD45, CD22; negative cytoplasmic IgM, surface IgM, CD117, Ig Kappa, Ig Lambda, CD15, CD65, NG2, CD21. In correlation with the Acute Leukemia Orientation Tube (ALOT) panel, a BCR-ABL1 Ph–positive B-cell acute lymphoblastic leukemia diagnosis was made [27]. The p190 BCR-ABL1 gene rearrangement was identified by reverse transcription polymerase chain reaction based on the international BIOMED-1 protocol (Figure 3f) [28]. 

## 3. Discussion

Only seven cases of hematologic malignancy occurrence post mRNA COVID-19 vaccinations were reported to date [7,8,9]. All of the seven cases received monovalent BNT162b2 (Pfizer-BioNTech) vaccine inoculations. Four of the seven cases presented acute myeloid leukemia, but their total number of COVID-19 vaccine doses received varied from two to five doses and the onset time of symptoms was about four to five weeks after the last vaccination. In our patient, the BCR-ABL1 Ph–positive (p190 form) B-cell acute lymphoblastic leukemia occurred just five days after the bivalent COVID-19 booster vaccine inoculation. To the best of our knowledge, this is the first case of Ph–positive B-cell acute lymphoblastic leukemia occurring after a bivalent mRNA COVID-19 vaccine booster. Adult acute lymphoblastic leukemia is a rare disease with a poor prognosis. The 5-year overall survival is only 35% in patients aged between 18 to 60 years [29]. The etiology of adult acute lymphoblastic leukemia includes old age (>70 years), recent viral infection, chemotherapy or radiation exposure, and genetic disorders [22]. Our patient was an otherwise healthy early middle-aged woman with no risk factors for acute lymphoblastic leukemia. Although she had a SARS-CoV-2 infection on 19 August 2022, neither clinical sequelae nor abnormal laboratory test results were investigated in the health examinations three months before this bivalent Moderna vaccine booster. The blood test and the abdominal sonography of the health examinations reported normal white blood cell count without blast, and no splenomegaly. Therefore, this case report might present a possible correlation between the development of Ph–positive B-cell acute lymphoblastic leukemia and bivalent mRNA vaccinations.

Immuno-pathological features have been frequently described after SARS-CoV-2 infection or COVID-19 vaccination. A 54-year-old man with a transient increase in blast count following COVID-19 infection mimicking acute leukemia has been reported [30]. He presented with some ailments of acute leukemia, such as petechiae, fever, and fatigue. His white blood cell count, hemoglobin, and platelet were 3.9 × 10^9^/L, 111 g/L, and 84 × 10^9^/L, respectively. In the differential count of white blood cell, neutrophil, lymphocyte, monocyte, and large unstained cells were 3.0%, 21.0%, 46.0%, and 30.0%, respectively. The blood smear exhibited a predominance of lymphocyte and monocyte, and 6% blasts. The bone marrow aspiration and biopsy showed hypocellular marrow with a predominant proliferation of monocytes and lymphocytes, and 23% blasts. In their report, the peripheral blood examinations and bone marrow studies improved within two weeks without further management. Our patient presented significantly different data from blood examinations and bone marrow studies from their case. A total of 90% of blasts were found in the peripheral blood of our patient. The bone marrow studies revealed hypercellular marrow with cellularity of more than 90%, and 68% blastic infiltration. Therefore, the standard therapies for BCR-ABL1 Ph–positive B-cell acute lymphoblastic leukemia, including dasatinib and chemotherapy agents, were administered. Her acute lymphoblastic leukemia got remission after hospitalization for approximately one month. 

Infection-neutralizing antibody responses constitute the major component of antiviral immunity. Wratil et al. have indicated consecutive spike antigen exposures, either by SARS-CoV-2 infection or COVID-19 vaccination, resulted in an increasing neutralization capacity [31]. Memory B cells, which are generated in the germinal centers, play an important role in long-term host defenses against viruses. These high-affinity memory B cells could persist up to 6 months after SARS-CoV-2 infection [32]. Compared with traditional vaccines, these new technology mRNA COVID-19 vaccines provided more efficient antigen-specific germinal center responses to produce memory B cells [33]. Our patient received a bivalent mRNA-1273 COVID-19 vaccine just 5 months after her SARS-CoV-2 infection. These repeated spike antigen exposures which amplified the immune cell response in a relatively short-term period might increase the incidence of B-cell acute lymphoblastic leukemia. A preliminary study mentioned the influence of SARS-CoV-2 spike protein on hematopoiesis and myeloid differentiation ex vivo [34]. Saluja et al. have reported a patient with COVID-19 pneumonia with subsequent chronic lymphocystic leukemia development [35]. The authors stated that SARS-CoV-2 infection stimulates an intense immune response with an elevation of pro-inflammatory cytokines, including interleukin-1, interleukin-6, interleukin-8, and tumor necrosis factor-α. We therefore propose that the anti-spike protein immune responses following a total of six spike antigen exposures in 1.5 years could be the trigger for Ph–positive B-cell ALL in our patient.

## 4. Conclusions

In conclusion, we presented the first case of Ph–positive B-cell acute lymphoblastic leukemia occurring five days after a booster dose of the bivalent mRNA COVID-19 vaccine. Although valuable pre-vaccine test results and comprehensive bone marrow studies were provided, we cannot conclude the causal relationship between bivalent vaccinations and the subsequent occurrence of Ph–positive B-cell acute lymphoblastic leukemia. Robust population-level studies would be required to determine whether there is an increased incidence of hematolymphoid neoplasms following vaccination. It is imperative to keep monitoring the hematopoietic adverse events after these new technology bivalent mRNA COVID-19 vaccinations, especially for patients with multiple spike antigen exposures in a relatively short-term period. Further pre-clinical studies for the safety evaluation of these vaccines are required.

## Figures and Tables

**Figure 1 medicina-59-00627-f001:**
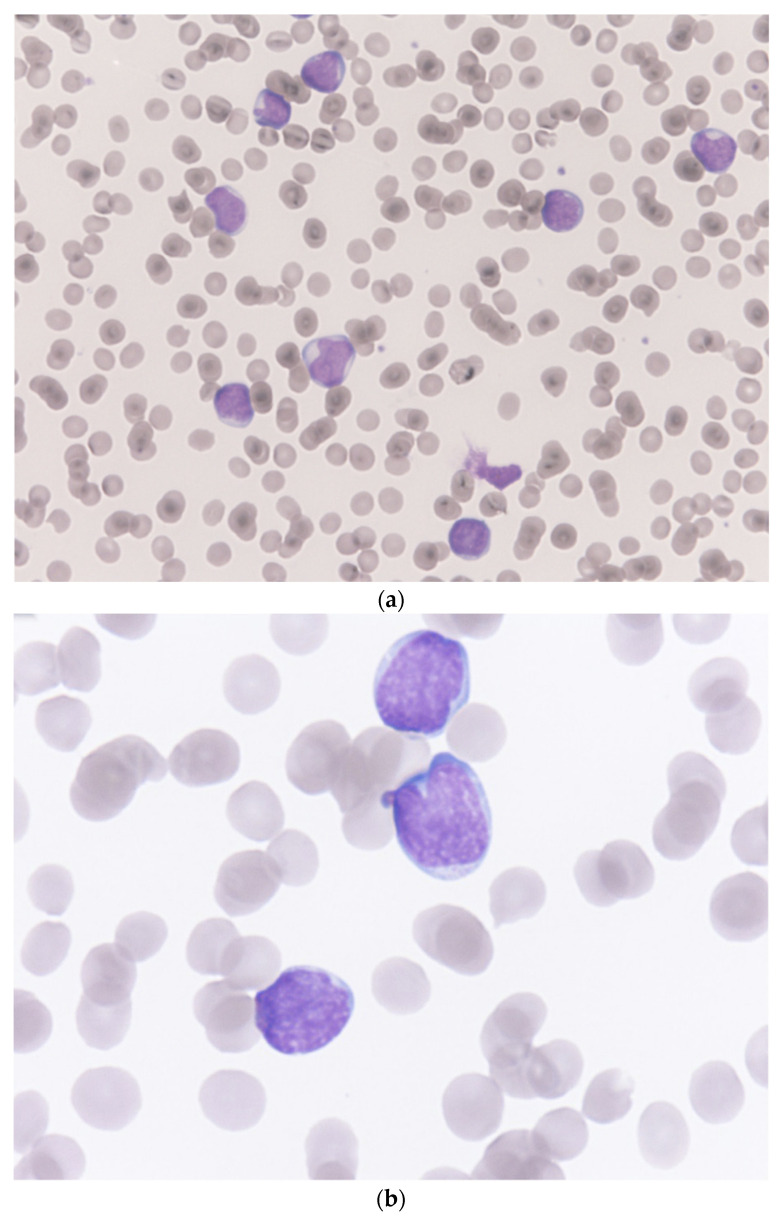
Microscopic pictures of a Giemsa–Wright stained peripheral blood smear from the patient. (**a**) almost all the white blood cells on the visual field were blasts (Magnification 400×). (**b**) these large cells with irregular/clefting nuclei, coarse, clumped chromatin, and occasional nucleoli, and high nuclear-to-cytoplasmic ratio indicated acute lymphoblastic leukemia (Magnification 1000×).

**Figure 2 medicina-59-00627-f002:**
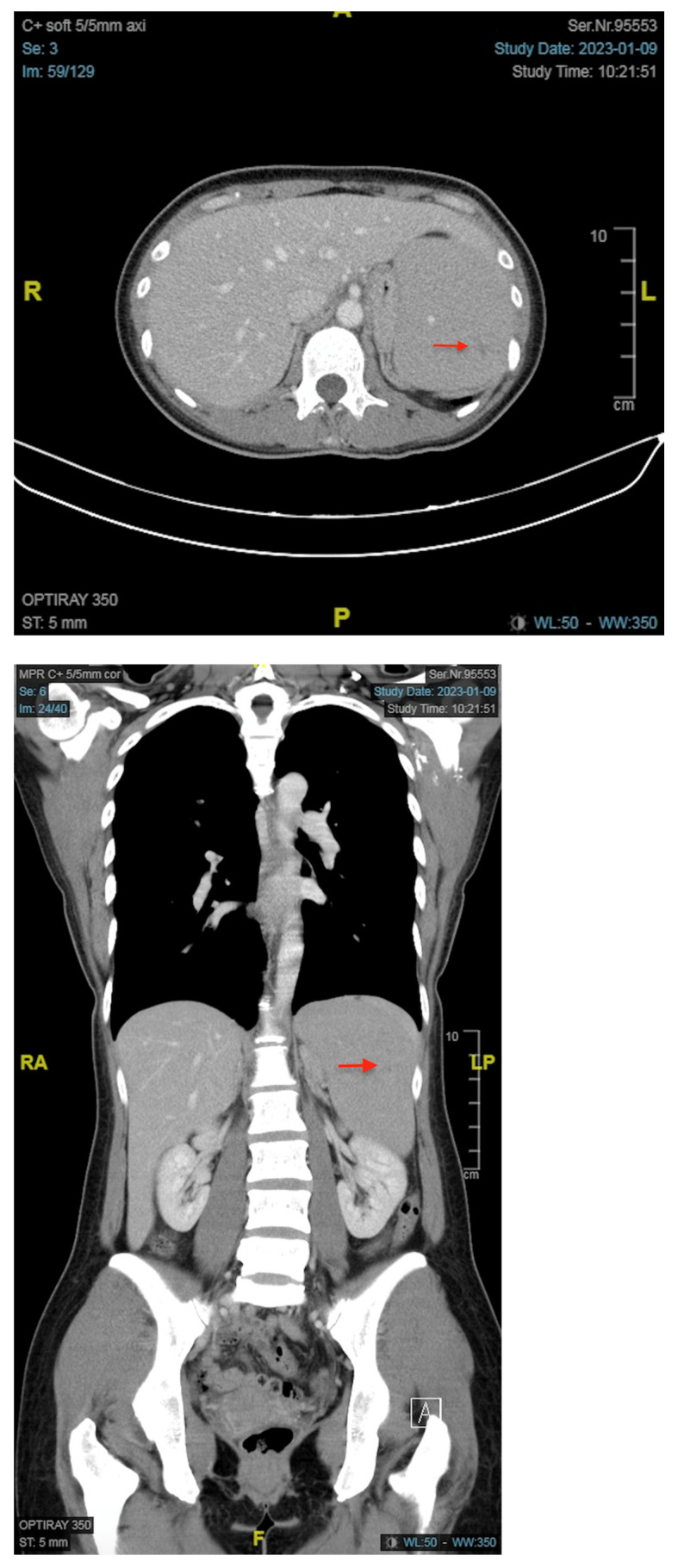
Contrast-enhanced abdominal computed tomography images of the patient. Both the axial and coronal views demonstrated splenomegaly with a tiny splenic infarct (arrow).

**Figure 3 medicina-59-00627-f003:**
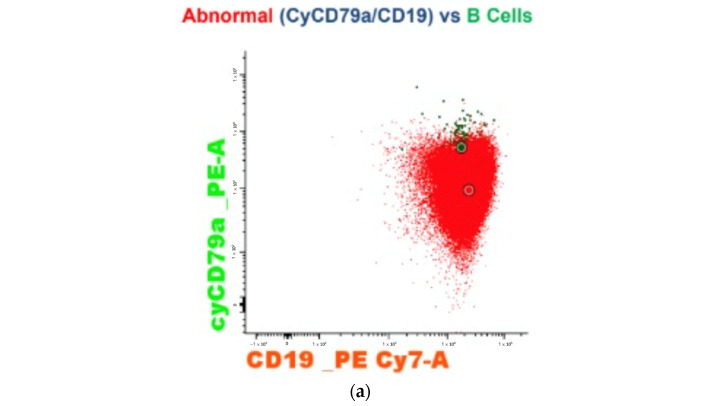
Genetic subtype study of the patient. (**a**,**b**) the immunophenotype characterization with the use of flow cytometry demonstrated moderate cytoplasmic CD79a, CD19, and CD34 expression. (**c**–**e**) in B-cell precursor acute lymphoblastic leukemia diagnostic panel, the specimen showed brightness of CD58 and CD10 with moderate expression of TdT. (**f**) reverse transcription polymerase chain reaction revealed significant expression of the BCR-ABL1 gene (p190 form).

## Data Availability

The data presented in this study are available on request from the corresponding author.

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
