# Peer review of "Ph-Positive B-Cell Acute Lymphoblastic Leukemia Occurring after Receipt of Bivalent SARS-CoV-2 mRNA Vaccine Booster: A Case Report"

_medicina, 2023, doi:10.3390/medicina59030627_

Round 1

Reviewer 1 Report

The manuscript entitled “Ph-Positive B-Cell Acute Lymphoblastic Leukemia Occurring After Receipt of Bivalent SARS-CoV-2 mRNA Vaccine Booster: A Case Reporte" by Shy-Yau Ang et al. present the first case of a patient who developed Ph-positive B-cell acute lymphoblastic leukemia (ALL) after five days of receiving a COVID-19 booster vaccination with mRNA-1273 (Moderna). The authors of this case report speculate a possible correlation between the bivalent vaccinations and the subsequent development of Ph–positive B-cell ALL, in which the anti-spike protein immune responses could be a trigger for leukemia. This study sheds light on the major safety concerns about the bivalent mRNA COVID-19 vaccine. The study is an interesting and important topic.

1)      The authors should write at least one paragraph in the result section to describe the test and the analysis in more detail as well as provide more details about the equipment used in this case report, especially the microscope used to get on figure 1 as well as the Realtime polymerase chain reaction of figure 2 f.

2)      Figure 2(a-e) should be improved and provide good-resolution figures in which the axes are clearly visible. Figure 2 f should be updated with a higher resolution figure and better labeling and avoid using the Gride inside the figure.

3)      There are many typos. For example, change the case reporte in line 4 to the case report, omicron in line 37 to Omicron, etc.

Author Response

Response to Reviewer 3 Comments Point 1: The authors should write at least one paragraph in the result section to describe the test and the analysis in more detail as well as provide more details about the equipment used in this case report, especially the microscope used to get on figure 1 as well as the Realtime polymerase chain reaction of figure 2 f. Response 1: Thanks for your constructive comment. The result section was revised as follows. (Page 6-7) The microscopic examination of a Giemsa-Wright stained peripheral blood smear revealed almost all the white blood cells on the visual field were blasts and these large white blood cells presented with irregular/clefting nuclei, coarse, clumped chromatin, and occasional nucleoli and high nuclear-to-cytoplasmic ratio (Figure 1). The results of C-reactive protein, lactate, and LDH were 1.455 mg/dL, 2.28 mmol/L, and 240 U/L, respectively. Two days later, the bone marrow biopsy and aspiration studies were conducted. A 1.2 x 0.2 x 0.2 cm brown, hard and core-like bone marrow specimen was obtained from the left iliac crest of the patient. The specimen was fixed in formalin subsequently. Microscopically, it shows hypercellular marrow with cellularity more than 90%. Most marrow spaces and hematopoietic components are replaced by small to medium sized primitive round blue cells. Immunohistochemically, the lesion cells are positive for CD34 and Terminal deoxynucleotidyl transferase (TdT); negative for CD117 and myeloperoxidase. CD3 shows only scattered positivity, and CD20 stains on more cells (about 20% of the nucleated cells) than CD3. The bone marrow aspiration showed 68% blastic infiltration. The immunophenotype characterization with the use of flow cytometry demonstrated moderate cytoplasmic CD79a (Figure 2a), CD19 (Figure 2a), and CD34 (Figure 2b) expression with negative cytoplasmic myeloperoxidase and CD3. It was compatible with B-cell precursor acute lymphoblastic leukemia. In B-cell precursor acute lymphoblastic leukemia diagnostic panel, the specimen showed brightness of CD58 (Figure 2c) and CD10 (Figure 2d) with moderate expression of CD34, CD19, CD66c, CD38, TdT (Figure 2e), CD24; dim CD45, CD22; negative cytoplasmic IgM, surface IgM, CD117, Ig Kappa, Ig Lambda, CD15, CD65, NG2, CD21. In correlation with the Acute Leukemia Orientation Tube (ALOT) panel, a BCR-ABL1 Ph–positive B-cell acute lymphoblastic leukemia diagnosis was made [27]. The p190 BCR-ABL1 gene rearrangement was identified by reverse transcription polymerase chain reaction based on the international BIOMED-1 protocol (Figure 2f) [28]. References: [27] Kulis J, Sędek Ł, Słota Ł, Perkowski B, Szczepański T. Commonly Assessed Markers in Childhood BCP-ALL Diagnostic Panels and Their Association with Genetic Aberrations and Outcome Prediction. Genes (Basel). 2022;13. [28] van Dongen JJ, Macintyre EA, Gabert JA, Delabesse E, Rossi V, Saglio G, et al. Standardized RT-PCR analysis of fusion gene transcripts from chromosome aberrations in acute leukemia for detection of minimal residual disease. Report of the BIOMED-1 Concerted Action: investigation of minimal residual disease in acute leukemia. Leukemia. 1999;13:1901-28. Point 2: Figure 2(a-e) should be improved and provide good-resolution figures in which the axes are clearly visible. Figure 2 f should be updated with a higher resolution figure and better labeling and avoid using the Gride inside the figure. Response 2: Thanks for your constructive suggestion. We have done our best to improve the resolution of Figure 2(a-e). Figure 2f was revised, too. Please see the revised manuscript. Point 3: There are many typos. For example, change the case reporte in line 4 to the case report, omicron in line 37 to Omicron, etc. Response 3: Thanks for your constructive comment. The typos have been corrected and we have checked the revised manuscript carefully.

Reviewer 2 Report

The idea of ​​the article is original and interesting, however, it is necessary to address several points to be considered for publication; they are:

- The introduction needs to create a narrative talking about the possible relationship of ALL with vaccines for COVID-19

- The methodology was not transparent in how you related the case to LLA.

- The discussion needs more elements to support the presented narrative. Because it seemed to me a little unconnected between the subjects addressed.

Accordingly, I strongly suggest a significant revision of your discussion and conclusion.

Author Response

Response to Reviewer 1 Comments

Point 1: The introduction needs to create a narrative talking about the possible relationship of ALL with vaccines for COVID-19

Response 1: Thanks for your constructive comment. The introduction was revised as follows.

(Page 3-4) In the era of the coronavirus disease 2019 (COVID-19) pandemic, more than 750 million people have been infected globally, with over 6 million associated death [1]. Herd immunity was recommended as one of the best resolutions for this severe acute respiratory syndrome coronavirus 2 (SARS-CoV-2) crisis. Several vaccines for preventing SARS-CoV-2 infection were developed, such as Oxford/AstraZeneca COVID-19 vaccine, Pfizer BioNTech (BNT162b2) vaccine, Moderna COVID-19 (mRNA-1273) vaccine, and Novavax (NVX-CoV2373) vaccine. Although they have shown the effectiveness in reducing COVID-19 related mortality and morbidity [2-5], many concerns about the safety of these vaccines were generated regarding their novel genetic technology and materials [6]. Sporadic hematopoietic adverse events were reported [7-10]. The bivalent formulations of BNT162b2 (Pfizer-BioNTech) and mRNA-1273 (Moderna) COVID-19 vaccines, which can induce remarkable high levels of anti-spike protein antibodies, have demonstrated their significant efficacy against the Omicron variant and acceptable safety in clinical practice [11]. They are recommended as the first choice of COVID-19 vaccine boosters to protect against the infection of SARS-CoV-2 variants worldwide.

Acute lymphoblastic leukemia is a hematologic malignancy with heterogeneous cytogenetic appearance and prognosis [12]. It is an uncommon disease with low incidence rate about 1.7 per 100,000 persons per year in the United States [13]. The genetic diseases, such as Fanconi anemia [14], Klinefelter syndrome [15], Bloom syndrome [16], neurofibromatosis [17], ataxia-telangiectasia [18], and Down syndrome [19] were reported to be contributed to the development of acute lymphoblastic leukemia. Other risk factors of acute lymphoblastic leukemia include old age (>70 years), chemotherapy, radiation exposure, Epstein-Barr virus, and human immunodeficiency virus infection [20-22]. Although tyrosine kinase inhibitors targeted therapy significantly improved the outcome of patients with Philadelphia (Ph) chromosome–positive acute lymphoblastic leukemia, poor prognosis was reported in patients with relapsed and refractory Ph-positive acute lymphoblastic leukemia [23].

Many researchers have provided strong support for the safety and benefits of COVID-19 vaccination [2-5, 11, 24], but several authors implied immune cell stimulation might be the trigger of the hematologic adverse events after COVID-19 vaccination [7-10, 25, 26]. Intense vaccination campaign for herd immunity to against SARS-CoV-2 infection is mandatory, however, it is a continuous process for collecting the safety data of these vaccines, especially the current worldwide used booster bivalent formulations of BNT162b2 (Pfizer-BioNTech) and mRNA-1273 (Moderna) COVID-19 vaccines. Even if such sporadic events can happen coincidentally, they should be reported to shed light on the major safety concerns about these bivalent mRNA COVID-19 vaccines.

References:

[1] WHO Coronavirus (COVID-19) Dashboard.

[2] Polack FP, Thomas SJ, Kitchin N, Absalon J, Gurtman A, Lockhart S, et al. Safety and Efficacy of the BNT162b2 mRNA Covid-19 Vaccine. The New England journal of medicine. 2020;383:2603-15.

[3] Baden LR, El Sahly HM, Essink B, Kotloff K, Frey S, Novak R, et al. Efficacy and Safety of the mRNA-1273 SARS-CoV-2 Vaccine. The New England journal of medicine. 2021;384:403-16.

[4] Heath PT, Galiza EP, Baxter DN, Boffito M, Browne D, Burns F, et al. Safety and Efficacy of NVX-CoV2373 Covid-19 Vaccine. The New England journal of medicine. 2021;385:1172-83.

[5] Voysey M, Clemens SAC, Madhi SA, Weckx LY, Folegatti PM, Aley PK, et al. Safety and efficacy of the ChAdOx1 nCoV-19 vaccine (AZD1222) against SARS-CoV-2: an interim analysis of four randomised controlled trials in Brazil, South Africa, and the UK. Lancet (London, England). 2021;397:99-111.

[6] Solís Arce JS, Warren SS, Meriggi NF, Scacco A, McMurry N, Voors M, et al. COVID-19 vaccine acceptance and hesitancy in low- and middle-income countries. Nat Med. 2021;27:1385-94.

[7] Çınar OE, Erdoğdu B, Karadeniz M, Ünal S, Malkan Ü Y, Göker H, et al. Comment on Zamfir et al. Hematologic Malignancies Diagnosed in the Context of the mRNA COVID-19 Vaccination Campaign: A Report of Two Cases. Medicina 2022, 58, 874. Medicina (Kaunas). 2022;58.

[8] Erdogdu B, Çınar OE, Malkan Ü Y, Aksu S, Demiroglu H, Buyukasik Y, et al. Hematopoietic Adverse Events Associated with BNT162b2 mRNA Covid-19 Vaccine. International Journal of Hematology and Oncology. 2022;32:65-7.

[9] Zamfir MA, Moraru L, Dobrea C, Scheau AE, Iacob S, Moldovan C, et al. Hematologic Malignancies Diagnosed in the Context of the mRNA COVID-19 Vaccination Campaign: A Report of Two Cases. Medicina (Kaunas). 2022;58.

[10] Chen CY, Chen TT, Hsieh CY, Lien MY, Yeh SP, Chen CC. Case reports of management of aplastic anemia after COVID-19 vaccination: a single institute experience in Taiwan. Int J Hematol. 2023;117:149-52.

[11] Hause AM, Marquez P, Zhang B, Myers TR, Gee J, Su JR, et al. Safety Monitoring of Bivalent COVID-19 mRNA Vaccine Booster Doses Among Persons Aged ≥12 Years - United States, August 31-October 23, 2022. MMWR Morb Mortal Wkly Rep. 2022;71:1401-6.

[12] Paul S, Kantarjian H, Jabbour EJ. Adult Acute Lymphoblastic Leukemia. Mayo Clin Proc. 2016;91:1645-66.

[13] Pui CH, Robison LL, Look AT. Acute lymphoblastic leukaemia. Lancet (London, England). 2008;371:1030-43.

[14] Alter BP. Fanconi anemia and the development of leukemia. Best Pract Res Clin Haematol. 2014;27:214-21.

[15] Shaw MP, Eden OB, Grace E, Ellis PM. Acute lymphoblastic leukemia and Klinefelter's syndrome. Pediatr Hematol Oncol. 1992;9:81-5.

[16] Passarge E. Bloom's syndrome: the German experience. Ann Genet. 1991;34:179-97.

[17] Stiller CA, Chessells JM, Fitchett M. Neurofibromatosis and childhood leukaemia/lymphoma: a population-based UKCCSG study. Br J Cancer. 1994;70:969-72.

[18] Taylor AM, Metcalfe JA, Thick J, Mak YF. Leukemia and lymphoma in ataxia telangiectasia. Blood. 1996;87:423-38.

[19] Whitlock JA. Down syndrome and acute lymphoblastic leukaemia. Br J Haematol. 2006;135:595-602.

[20] Gérinière L, Bastion Y, Dumontet C, Salles G, Espinouse D, Coiffier B. Heterogeneity of acute lymphoblastic leukemia in HIV-seropositive patients. Annals of oncology : official journal of the European Society for Medical Oncology / ESMO. 1994;5:437-40.

[21] Sakajiri S, Mori K, Isobe Y, Kawamata N, Oshimi K. Epstein-Barr virus-associated T-cell acute lymphoblastic leukaemia. Br J Haematol. 2002;117:127-9.

[22] Alvarnas JC, Brown PA, Aoun P, Ballen KK, Barta SK, Borate U, et al. Acute Lymphoblastic Leukemia, Version 2.2015. Journal of the National Comprehensive Cancer Network : JNCCN. 2015;13:1240-79.

[23] Park HS. Current treatment strategies for Philadelphia chromosome-positive adult acute lymphoblastic leukemia. Blood Res. 2020;55:S32-s6.

[24] Shulman RM, Weinberg DS, Ross EA, Ruth K, Rall GF, Olszanski AJ, et al. Adverse Events Reported by Patients With Cancer After Administration of a 2-Dose mRNA COVID-19 Vaccine. Journal of the National Comprehensive Cancer Network : JNCCN. 2022;20:160-6.

[25] Paulsen FO, Schaefers C, Langer F, Frenzel C, Wenzel U, Hengel FE, et al. Immune thrombocytopenic purpura after vaccination with COVID-19 vaccine (ChAdOx1 nCov-19). Blood. 2021;138:996-9.

[26] Murdych TM. A case of severe autoimmune hemolytic anemia after a receipt of a first dose of SARS-CoV-2 vaccine. Int J Lab Hematol. 2022;44:e10-e2.

Point 2: The methodology was not transparent in how you related the case to ALL.

Response 2: Thanks for your constructive comment. The methodology was revised as follows.

(Page 5-7) The patient received four COVID-19 vaccinations before the last injection, including two doses of adenoviral vector-based vaccines (Oxford/AstraZeneca), a half-dose of monovalent mRNA vaccine (Moderna), and a protein vaccine (NovaVax) on 2021/06/04, 2021/08/31, 2022/01/15, and 2022/07/15, respectively. There was no remarkable discomfort after these inoculations. She had a SARS-CoV-2 infection with only minimal symptoms on 2022/08/19 and recovered in two weeks without any sequelae. A general blood test, abdominal sonography, and low-dose chest computed tomography (CT) for health examination were done on 2022/10/12. The white blood cell count, red blood cell count, hemoglobin, hematocrit, MCV, MCH, MCHC, and platelet were 5,730/μL, 4,110,000/μL, 12.3 g/dL, 37.3%, 94.5 fL, 28.3 pg, 34.6 g/dL, and 202,400/μL, respectively. In the differential count of white blood cell, neutrophil, lymphocyte, monocyte, eosinophil, and basophil were 54.2%, 37.9%, 5.8%, 1.5%, and 0.6%, respectively. The abdominal sonography showed no splenomegaly.

At our emergency department, the white blood cell count, red blood cell count, hemoglobin, hematocrit, MCV, MCH, MCHC, and platelet were 46,390/μL, 2,530,000/μL, 6.8 g/dL, 20.9%, 82.6 fL, 26.9 pg, 32.5 g/dL, and 48,000/μL, respectively. In the differential count of white blood cell, neutrophil, lymphocyte, monocyte, eosinophil, basophil, and blast were 1.0%, 9.0%, 0.0%, 0.0%, 0.0%, and 90.0%, respectively. The microscopic examination of a Giemsa-Wright stained peripheral blood smear revealed almost all the white blood cells on the visual field were blasts and these large white blood cells presented with irregular/clefting nuclei, coarse, clumped chromatin, and occasional nucleoli and high nuclear-to-cytoplasmic ratio (Figure 1). The results of C-reactive protein, lactate, and LDH were 1.455 mg/dL, 2.28 mmol/L, and 240 U/L, respectively. Splenomegaly without enlargement of lymph nodes was shown on the contrast-enhanced CT of the abdomen.

Two days later, the bone marrow biopsy and aspiration studies were conducted. A 1.2 x 0.2 x 0.2 cm brown, hard and core-like bone marrow specimen was obtained from the left iliac crest of the patient. The specimen was fixed in formalin subsequently. Microscopically, it shows hypercellular marrow with cellularity more than 90%. Most marrow spaces and hematopoietic components are replaced by small to medium sized primitive round blue cells. Immunohistochemically, the lesion cells are positive for CD34 and Terminal deoxynucleotidyl transferase (TdT); negative for CD117 and myeloperoxidase. CD3 shows only scattered positivity, and CD20 stains on more cells (about 20% of the nucleated cells) than CD3.

The bone marrow aspiration showed 68% blastic infiltration. The immunophenotype characterization with the use of flow cytometry demonstrated moderate cytoplasmic CD79a (Figure 2a), CD19 (Figure 2a), and CD34 (Figure 2b) expression with negative cytoplasmic myeloperoxidase and CD3. It was compatible with B-cell precursor acute lymphoblastic leukemia. In B-cell precursor acute lymphoblastic leukemia diagnostic panel, the specimen showed brightness of CD58 (Figure 2c) and CD10 (Figure 2d) with moderate expression of CD34, CD19, CD66c, CD38, TdT (Figure 2e), CD24; dim CD45, CD22; negative cytoplasmic IgM, surface IgM, CD117, Ig Kappa, Ig Lambda, CD15, CD65, NG2, CD21. In correlation with the Acute Leukemia Orientation Tube (ALOT) panel, a BCR-ABL1 Ph–positive B-cell acute lymphoblastic leukemia diagnosis was made [27]. The p190 BCR-ABL1 gene rearrangement was identified by reverse transcription polymerase chain reaction based on the international BIOMED-1 protocol (Figure 2f) [28].

References:

[27] Kulis J, Sędek Ł, Słota Ł, Perkowski B, Szczepański T. Commonly Assessed Markers in Childhood BCP-ALL Diagnostic Panels and Their Association with Genetic Aberrations and Outcome Prediction. Genes (Basel). 2022;13.

[28] van Dongen JJ, Macintyre EA, Gabert JA, Delabesse E, Rossi V, Saglio G, et al. Standardized RT-PCR analysis of fusion gene transcripts from chromosome aberrations in acute leukemia for detection of minimal residual disease. Report of the BIOMED-1 Concerted Action: investigation of minimal residual disease in acute leukemia. Leukemia. 1999;13:1901-28.

Point 3: The discussion needs more elements to support the presented narrative. Because it seemed to me a little unconnected between the subjects addressed.

Response 3: Thanks for your constructive comment. The discussion has been revised as follows.

(Page 7-9) Only seven cases of hematologic malignancy occurrence post mRNA COVID-19 vaccinations were reported to date [7-9]. All of the seven cases received monovalent BNT162b2 (Pfizer-BioNTech) vaccine inoculations. Four of the seven cases presented acute myeloid leukemia, but their total number of COVID-19 vaccine doses received varied from two to five doses and the onset time of symptoms was about four to five weeks after the last vaccination. In our patient, the BCR-ABL1 Ph–positive (p190 form) B-cell acute lymphoblastic leukemia occurred just five days after the bivalent COVID-19 booster vaccine inoculation. To the best of our knowledge, this is the first case of Ph–positive B-cell acute lymphoblastic leukemia occurring after a bivalent mRNA COVID-19 vaccine booster. Adult acute lymphoblastic leukemia is a rare disease with poor prognosis. The 5-year overall survival is only 35% in patients age between 18 to 60 years [29]. The etiology of adult acute lymphoblastic leukemia includes old age (>70 years), recent viral infection, chemotherapy or radiation exposure, and genetic disorders [22]. Our patient was an otherwise healthy early middle-aged woman with no risk factors for acute lymphoblastic leukemia. Although she had a SARS-CoV-2 infection on 2022/08/19, neither clinical sequelae nor abnormal laboratory test results were investigated in the health examinations three months before this bivalent Moderna vaccine booster. The blood test and the abdominal sonography of the health examinations reported normal white blood cell count without blast, and no splenomegaly, respectively. Therefore, this case report might present a possible correlation between the development of Ph–positive B-cell acute lymphoblastic leukemia and bivalent mRNA vaccinations.

Infection-neutralizing antibody responses constitute the major component of antiviral immunity. Wratil et al. have indicated consecutive spike antigen exposures, either by SARS-CoV-2 infection or COVID-19 vaccination, resulted in an increasing neutralization capacity [30]. On the other hand, these spike antigen exposures which amplified the immune cell response might increase the incidence of hematologic adverse events. A preliminary study mentioned the influence of SARS-CoV-2 spike protein on hematopoiesis and myeloid differentiation ex vivo [31]. Saluja et al. have reported a patient of COVID-19 pneumonia with subsequent chronic lymphocystic leukemia development [32]. The authors stated SARS-CoV-2infection stimulates an intense immune response with elevation of pro-inflammatory cytokines, including interleukin-1, interleukin-6, interleukin-8, and tumor necrosis factor-α. We therefore propose that the anti-spike protein immune responses following a total of six spike antigen exposures in 1.5 years could be the trigger for Ph–positive B-cell ALL in our patient.

References:

[29] Bassan R, Hoelzer D. Modern therapy of acute lymphoblastic leukemia. Journal of clinical oncology : official journal of the American Society of Clinical Oncology. 2011;29:532-43.

[30] Wratil PR, Stern M, Priller A, Willmann A, Almanzar G, Vogel E, et al. Three exposures to the spike protein of SARS-CoV-2 by either infection or vaccination elicit superior neutralizing immunity to all variants of concern. Nat Med. 2022;28:496-503.

[31] Ropa J, Cooper S, Capitano M, Broxmeyer H. Sars-Cov-2 Spike Protein Induces Cellular Changes in Primitive and Mature Hematopoietic Cells2020.

[32] Saluja P, Gautam N, Amisha F, Safar M, Bartter T. Emergence of Chronic Lymphocytic Leukemia During Admission for COVID-19: Cause or Coincidence? Cureus. 2022;14:e23470.

Point 4: Accordingly, I strongly suggest a significant revision of your discussion and conclusion.

Response 4: Thanks for your constructive comment. The conclusions has been revised as follows.

(Page 9) In conclusion, we present the first case of Ph–positive B-cell acute lymphoblastic leukemia occurring five days after a booster dose of the bivalent mRNA COVID-19 vaccine. Although valuable pre-vaccine test results and comprehensive bone marrow studies were provided, we cannot conclude the casual relationship between bivalent vaccinations and the subsequent occurrence of Ph–positive B-cell acute lymphoblastic leukemia. This case might be coincidental given the incredibly high population rates of vaccination and/or prior SARS-CoV-2 infection. Robust population-level studies would be required to determine whether there is an increased incidence of hematolymphoid neoplasms following vaccination. It is imperative to monitor the hematopoietic adverse events after bivalent mRNA COVID-19 vaccinations, especially for patients with multiple spike antigen exposures in a relatively short-term period. Further pre-clinical studies for the safety evaluation of these vaccines are required.

Reviewer 3 Report

As the authors agree the relationship between bivalent vaccinations and the subsequent occurrence of Ph–positive B-cell ALL cannot be concluded from this case report. However its imperative to track any such events.

I have the following comments/suggestions:

The tests for Ph–positive B-cell ALL were not conducted on the patient before vaccination, authors need to be careful before drawing any correlation. 

The images used in Figure 2 is blurred. Should be replaced with clear images.

The same figure legend has been used for each image of a figure. Instead the figure legend should be used once to describe all the images in a figure.

A little detail on the methods of obtaining these images should be mentioned.

Author Response

Response to Reviewer 2 Comments

Point 1: The tests for Ph–positive B-cell ALL were not conducted on the patient before vaccination, authors need to be careful before drawing any correlation.

Response 1: Thanks for your constructive comment. The method section was revised as follows.

(Page 5) A general blood test, abdominal sonography, and low-dose chest computed tomography (CT) for health examination were done on 2022/10/12. The white blood cell count, red blood cell count, hemoglobin, hematocrit, MCV, MCH, MCHC, and platelet were 5,730/μL, 4,110,000/μL, 12.3 g/dL, 37.3%, 94.5 fL, 28.3 pg, 34.6 g/dL, and 202,400/μL, respectively. In the differential count of white blood cell, neutrophil, lymphocyte, monocyte, eosinophil, and basophil were 54.2%, 37.9%, 5.8%, 1.5%, and 0.6%, respectively. The abdominal sonography showed no splenomegaly.

(Page 8) The blood test and the abdominal sonography of the health examinations reported normal white blood cell count without blast, and no splenomegaly. Therefore, this case report might present a possible correlation between the development of Ph–positive B-cell acute lymphoblastic leukemia and bivalent mRNA vaccinations.

Point 2: The images used in Figure 2 is blurred. Should be replaced with clear images.

Response 2: Thanks for your constructive comment. We have done our best to improve the resolution of Figure 2. Please see the revised manuscript.

Point 3: The same figure legend has been used for each image of a figure. Instead the figure legend should be used once to describe all the images in a figure.

Response 3: Thanks for your constructive comment. The figure legend was revised as follows.

Figure 1. Microscopic pictures of a Giemsa-Wright stained peripheral blood smear from the patient. (a) almost all the white blood cells on the visual field were blasts (Magnification 400X). (b) these large cells with irregular/clefting nuclei, coarse, clumped chromatin, and occasional nucleoli and high nuclear-to-cytoplasmic ratio indicated acute lymphoblastic leukemia (Magnification 1000X).

Figure 2. Genetic subtype study of the patient. (a)-(b) the immunophenotype characterization with the use of flow cytometry demonstrated moderate cytoplasmic CD79a, CD19, and CD34 expression. (c)-(e) in B-cell precursor acute lymphoblastic leukemia diagnostic panel, the specimen showed brightness of CD58 and CD10 with moderate expression of TdT. (f) reverse transcription polymerase chain reaction revealed significant expression of BCR-ABL1 gene (p190 form).

Point 4: A little detail on the methods of obtaining these images should be mentioned.

Response 4: Thanks for your constructive comment. The method section was revised as follows.

(Page 6-7) The microscopic examination of a Giemsa-Wright stained peripheral blood smear revealed almost all the white blood cells on the visual field were blasts and these large white blood cells presented with irregular/clefting nuclei, coarse, clumped chromatin, and occasional nucleoli and high nuclear-to-cytoplasmic ratio (Figure 1). The results of C-reactive protein, lactate, and LDH were 1.455 mg/dL, 2.28 mmol/L, and 240 U/L, respectively. Splenomegaly without enlargement of lymph nodes was shown on the contrast-enhanced CT of the abdomen.

Two days later, the bone marrow biopsy and aspiration studies were conducted. A 1.2 x 0.2 x 0.2 cm brown, hard and core-like bone marrow specimen was obtained from the left iliac crest of the patient. The specimen was fixed in formalin subsequently. Microscopically, it shows hypercellular marrow with cellularity more than 90%. Most marrow spaces and hematopoietic components are replaced by small to medium sized primitive round blue cells. Immunohistochemically, the lesion cells are positive for CD34 and Terminal deoxynucleotidyl transferase (TdT); negative for CD117 and myeloperoxidase. CD3 shows only scattered positivity, and CD20 stains on more cells (about 20% of the nucleated cells) than CD3.

The bone marrow aspiration showed 68% blastic infiltration. The immunophenotype characterization with the use of flow cytometry demonstrated moderate cytoplasmic CD79a (Figure 2a), CD19 (Figure 2a), and CD34 (Figure 2b) expression with negative cytoplasmic myeloperoxidase and CD3. It was compatible with B-cell precursor acute lymphoblastic leukemia. In B-cell precursor acute lymphoblastic leukemia diagnostic panel, the specimen showed brightness of CD58 (Figure 2c) and CD10 (Figure 2d) with moderate expression of CD34, CD19, CD66c, CD38, TdT (Figure 2e), CD24; dim CD45, CD22; negative cytoplasmic IgM, surface IgM, CD117, Ig Kappa, Ig Lambda, CD15, CD65, NG2, CD21. In correlation with the Acute Leukemia Orientation Tube (ALOT) panel, a BCR-ABL1 Ph–positive B-cell acute lymphoblastic leukemia diagnosis was made [27]. The p190 BCR-ABL1 gene rearrangement was identified by reverse transcription polymerase chain reaction based on the international BIOMED-1 protocol (Figure 2f) [28].

References:

[27] Kulis J, Sędek Ł, Słota Ł, Perkowski B, Szczepański T. Commonly Assessed Markers in Childhood BCP-ALL Diagnostic Panels and Their Association with Genetic Aberrations and Outcome Prediction. Genes (Basel). 2022;13.

[28] van Dongen JJ, Macintyre EA, Gabert JA, Delabesse E, Rossi V, Saglio G, et al. Standardized RT-PCR analysis of fusion gene transcripts from chromosome aberrations in acute leukemia for detection of minimal residual disease. Report of the BIOMED-1 Concerted Action: investigation of minimal residual disease in acute leukemia. Leukemia. 1999;13:1901-28.

Round 2

Reviewer 2 Report

Line 79 - It would be interesting to insert the CT

The attempt to establish a relationship between vaccination and the reported case is still weak. How did you arrive at this relationship?

Your conclusion needed to be more straightforward. Is there a possible relationship between the vaccination and the case that occurred?

Author Response

Response to Reviewer 2 Comments

Point 1: Line 79 - It would be interesting to insert the CT

Response 1: Thanks for your constructive suggestion. The CT images were inserted as Figure 2.

(Page 6) Splenomegaly with tiny splenic infarct, and no enlargement of lymph nodes were shown on the contrast-enhanced CT of the abdomen (Figure 2).

(a)(b)

Figure 2. Contrast enhanced abdominal computed tomography images of the patient. Both of the axial and coronal views demonstrated splenomegaly with tiny splenic infarct (arrow).

Point 2: The attempt to establish a relationship between vaccination and the reported case is still weak. How did you arrive at this relationship?

Response 2: Thanks for your constructive comment. We have done our best to improve the relationship between vaccination and the reported case carefully. Please see the revised manuscript.

(Page 8-9) Infection-neutralizing antibody responses constitute the major component of antiviral immunity. Wratil et al. have indicated consecutive spike antigen exposures, either by SARS-CoV-2 infection or COVID-19 vaccination, resulted in an increasing neutralization capacity [30]. Memory B cells, which are generated in the germinal centers, play an important role in long-term host defenses against viruses. These high-affinity memory B cells could persist up to 6 months after SARS-CoV-2 infection [31]. Compared with traditional vaccines, these new technology mRNA COVID-19 vaccines provided more efficient antigen-specific germinal center responses to produce memory B cells [32]. Our patient received a bivalent mRNA-1273 COVID-19 vaccine just 5 months after her SARS-CoV-2 infection. These repeated spike antigen exposures which amplified the immune cell response in a relatively short-term period might increase the incidence of B-cell acute lymphoblastic leukemia. A preliminary study mentioned the influence of SARS-CoV-2 spike protein on hematopoiesis and myeloid differentiation ex vivo [33]. Saluja et al. have reported a patient of COVID-19 pneumonia with subsequent chronic lymphocystic leukemia development [34]. The authors stated SARS-CoV-2 infection stimulates an intense immune response with elevation of pro-inflammatory cytokines, including interleukin-1, interleukin-6, interleukin-8, and tumor necrosis factor-α. We therefore propose that the anti-spike protein immune responses following a total of six spike antigen exposures in 1.5 years could be the trigger for Ph–positive B-cell ALL in our patient.

References:

  1. Sokal A, Chappert P, Barba-Spaeth G, Roeser A, Fourati S, Azzaoui I, et al. Maturation and persistence of the anti-SARS-CoV-2 memory B cell response. Cell. 2021;184:1201-1213.e14.
  2. Lederer K, Castaño D, Gómez Atria D, Oguin TH, 3rd, Wang S, Manzoni TB, et al. SARS-CoV-2 mRNA Vaccines Foster Potent Antigen-Specific Germinal Center Responses Associated with Neutralizing Antibody Generation. Immunity. 2020;53:1281-1295.e5.

Point 3: Your conclusion needed to be more straightforward. Is there a possible relationship between the vaccination and the case that occurred?

Response 3: Thanks for your constructive comment. The conclusions have been revised as follows.

(Page 9) In conclusion, we present the first case of Ph–positive B-cell acute lymphoblastic leukemia occurring five days after a booster dose of the bivalent mRNA COVID-19 vaccine. Although valuable pre-vaccine test results and comprehensive bone marrow studies were provided, we cannot conclude the casual relationship between bivalent vaccinations and the subsequent occurrence of Ph–positive B-cell acute lymphoblastic leukemia. Robust population-level studies would be required to determine whether there is an increased incidence of hematolymphoid neoplasms following vaccination. It is imperative to keep monitoring the hematopoietic adverse events after these new technology bivalent mRNA COVID-19 vaccinations, especially for patients with multiple spike antigen exposures in a relatively short-term period. Further pre-clinical studies for the safety evaluation of these vaccines are required.
